# A New Approach to the Structure-Properties Relationships Determination for Porous Filled Reinforced Materials

**DOI:** 10.3390/polym14204390

**Published:** 2022-10-18

**Authors:** Miroslav Černý, Josef Petruš, Veronika Pavliňáková

**Affiliations:** 1Faculty of Chemistry, Institute of Materials Science, Brno University of Technology, Purkyňova 464/118, 612 00 Brno, Czech Republic; 2Central European Institute of Technology, Brno University of Technology, Purkyňova 656/123, 612 00 Brno, Czech Republic

**Keywords:** reinforcement, composite, porosity, tensile modulus, tensile strength

## Abstract

This study describes a new mathematical approach to the relationship between mechanical properties (tensile modulus, ultimate strength, and strain), composition as well as structure of porous-filled reinforced composites. The composite system consisted of a polyurethane matrix, a rubber filler, and a small amount of polyethylene terephthalate as a reinforcement. The newly proposed equations are based on a special mixing rule with the same basic form for all studied properties. The mixing rule contains a correction parameter η, which differs in different filler content in the filled part of the composite. Here, a cubic exponential function including the product of suitable structural parameters and exponents ensuring the best fitting and describable by matrix properties were successfully defined to fit the different values of correction parameter. The proposed equations should be a suitable step to obtain a relationship for describing the mechanical behavior of porous-filled and reinforced composites in the case of a small amount of reinforcement.

## 1. Introduction

Filled porous composites are widely represented in our area as concrete or various types of mineral casting [1,2]. The same can be said for reinforced composites. Combinations of these can also be found, for example, as reinforced concrete, which contains a cement matrix, sand as filler, and steel as reinforcement, as well as voids [3]. Each material has its own mechanical behavior depending on its composition and structure. It could be very interesting (and perhaps very useful) to relate the composition and structure of a material with a random degree of complexity and its mechanical behavior through some relationships. This post is a sincere effort to contribute to that goal.

First, it is necessary to start with less complicated structures, such as simply porous materials, filled non-porous and porous composites, and follow the increasing complexity of the relationship(s) in this direction. The relationships for single-component porous materials can acquire various mathematical forms—linear, exponential, power [4,5], and even logarithmic [5]. Linear shapes are applicable only to materials with a relatively small number of pores [4]. The logarithmic approach is not suitable due to the limiting impossibility of the logarithmic base to reach negative values (experienced from the early time of our research), and this is also rare in the literature. Exponential and power functions are the most appropriate because the current power(s) can reach various positive or negative values and it is then easier to find some meaning for that power(s). In this work, all power and exponential relations are included under the term exponential, because it is assumed that the power type of the equation can also be exponential depending on the choice of what the variable is (whether it is an exponent or a base). Therefore, this work is focused on exponential functions due to their usable variability. The most common property to describe the elastic modulus is found in several similar exponential equations. Equation (1) is quite common for calculating the elastic modulus.
(1)$$E=E_{m}\cdot {\left(1-a\cdot n\right)}^{p}$$
where *E* and *E_m_* are the elastic modulus for the porous and non-porous material and *n* is the porosity. The description of parameters *a* and *p* may vary. The introduction of *a* as “packing geometry factor” and *p* as a term of the equation depending on the material grain morphology and pore geometry is mentioned in [6] or simply as material constants in [7]. Equation (1) is mentioned in research on ceramics [8,9,10], expanded graphite [6], and porous thermosets [7]. The same equation as (1) except for the *a* term can also be found in [11]. The second widely cited exponential equation for *E* and the porosity of one-component materials is in Equation (2).
(2)$$E=E_{m}\cdot e^{-b\cdot n}$$

This equation occurs commonly with Equation (1) in [6,8,9] with the caveat that the parameter *b* depends on the nature of the porosity. Moreover, it is used in research [11,12,13] dedicated to ceramics. Another Equation (3) with the terms belonging to density (*ρ*) and critical density (*ρ_c_*) is present in the works devoted to porous metals (mainly titanium alloys) [14,15] or metallic foams [16].
(3)$$\frac{E}{E_{m}}=a\cdot {\left(\frac{\rho }{\rho_{c}}\right)}^{b}$$

This could be rewritten in the form of Equation (4) using the porosity term.
(4)$$E=E_{m}\cdot a\cdot {\left(1-n\right)}^{b}$$

In addition to Equations (1) and (2), Equation (5) additionally contains the critical porosity *n_c_* [11].
(5)$$E=E_{m}\cdot {\left(1-\frac{n}{n_{c}}\right)}^{\frac{1}{J}}$$
where the values of *n_c_* and *J* were chosen to fit the data exactly. The basis of our proposed system is Equation (6) [10,17].
(6)$$E=E_{m}\cdot {\left(1-n\right)}^{b}$$

In addition to the elastic modulus, there is also an attempt to describe the strength of porous materials with similar relationships. For example, Equation (2) valid for elastic modulus has a twin for strength, Equation (7).
(7)$$\sigma =\sigma_{m}\cdot e^{–b\cdot n}$$
where values of *σ* and *σ_m_* are strength for porous and non-porous material [18]. The *b* parameter is equal approximately to 7 and is, according to authors, independent of material type. The articles [14,15] offer the equation as having the same shape as Equation (3) for *E* as for strength, Equation (8).
(8)$$\frac{\sigma }{\sigma_{m}}=a\cdot {\left(\frac{\rho }{\rho_{c}}\right)}^{b}$$

Another Equation (9) similar to Equation (8) and corresponding to Equation (6) valid for *E* is part of the basis of our system of relations. The equation is also present in [19] devoted to foamed concrete and in [4] serving as a review article.
(9)$$\sigma =\sigma_{m}\cdot {\left(1-n\right)}^{b}$$

The description of the mechanical behavior of a composite material (filled or reinforced) must generally be more complex due to more complex structure, even if these materials are porous. There are two approaches—microscopic based on knowledge of the microstructure, models, and/or numerical simulations, and usually, a simplified macroscopic approach based, for example, on the mixture rules forming an average of the behavior of material from the content and properties of its components [20]. The macroscopic approach is simpler in mathematical complexity. This dual path is also evident for single-component porous materials in Equations (1)–(9), as relations are shown here with an effort to incorporate the microstructure, for example, into the shape of pores as well as empirical relations obtained from experimental data. The first approach (microscopic) dominates in the field of porous composites [21,22,23,24,25,26,27,28,29]. It is usually based on various models [21,26,27,28,29,30] including knowledge of the microstructure or its assumption or simplification, numerical modeling, and various mathematical methods including fractals [21], finite elements method [23,24], and fast Fourier Transform [25]. Even the mixture rule is often a part or basis of various models [21,23,27,30]. Assumption or simplification of the microstructure, according to the literature, appears essential. The microstructure approach naturally involves many structural assumptions, such as the pore, particle, or fiber shapes or the means of contact between phases. Regarding the limits of the microstructure, scientific publications are not uniform. There are a number of differences. Among them can be included no touch of particles and voids [21] or fibers and voids [23], spherical void shape [29], granular shape of particles [22], direction and circular cross-section of nanofibers combined with the presence of nanopores [25], cylindrical shape of fibers with no contact with voids [23], discretization to the domain containing different particles from their whole distribution [21], filling by nanofibers or nanotubes [25,27,29], two-phase system, where the second phase is on the surface of the first and separates the first from open porosity [24] and layered character of composite [26].

There are also differences in observed mechanical properties, which mainly concern different types of elastic modulus or more precisely the elastic region of the material loading [21,23,25,29,30]. It is also an important limitation of most works in the literature. However, there is some effort to predict nonlinear stress [22,24,26,27,28], but mostly with above limits or load range limits. Limits include nonlinear stress only for a granular porous composite [22] or a simple 2-phase structure [24], the effect of thermal failure on the mechanical behavior of a layered structure [26], yield strength and ultimate strength/strain and absorbed energy, but only in case of nanofibrous filling [27] and yield strength of filled composite [28].

The microscopic approach is based on microstructural knowledge or assumptions of physics and mechanics, including the above-mentioned limits. For example, the stepwise averaging modeling based on a mixture rule with sequential assumptions incorporation is presented in [30]. Generally, this means that relationships (models) are first created and then applied or subjected to numerical modeling and then compared to selected experimental data with a higher or lower agreement. Instead, in this work, a macroscopic approach was chosen in order to be less limited by the microstructure of the material or studied properties. We assumed the homogeneity of the examined materials from the macroscopic point of view, except for the reinforcement. This way of work also allows more properties to be examined in the area than the elastic modulus, but also, for example, ultimate strength and strain.

This approach has also been used in our previous studies [31,32] for filled porous composites containing randomly shaped particles and pores that are randomly connected. An example showing the random structure typical of our material in a microscopic figure can be seen in [32]. The figure shows a material containing a white matrix contrasting with voids and black particles, which is not used in this work but exhibits the same or very similar structure to materials in this work. The macroscopic point of view may not be as accurate in the physical description of mechanical behavior as model utilization, but it has the great advantage of using quite easy mathematics (powers, logarithms). It is based on structural parameters reflecting the macroscopic composition of the porous material. However, porosity is a low convenient term for more complex materials than one component and in our case must be replaced by another structural parameter. Our previous work [31] studied porous composites and defined a new structural parameter called interspace filling, which defines how much of the volume lying between rubber particles is filled by the matrix. The interspace filling (*n_p_*) can be calculated using Equation (10).
(10)$$n_{p}=\left(1-\frac{n}{n+v_{m}}\right)=\left(1-\frac{n}{n+\frac{v_{m(t)}}{1+\frac{n}{1-n}}}\right)$$
where *v_m_* and *v_m(t)_* are the volume fractions of the matrix in the material differing by porosity inclusion and neglect. The relationships between composition and structure from a macroscopic point of view and several mechanical properties obtained by tensile testing, including tensile modulus, ultimate strength (*σ_Fmax_*) and strain (*ε_Fmax_*) and energy need to achieve ultimate strength (*A_Fmax_*) applicable for porous filled composites are described in [32]. This paper contains the interspace volume (1 − *v_f_*) as another applicable structural parameter. The interspace volume is a dimensionless parameter that can be calculated by Equation (11). The symbol *v_f_* in 1 − *v_f_* stands for the volume fraction of filler.
(11)$$1-v_{f}=n+v_{m}=n+n\cdot \frac{n_{p}}{1-n_{p}}=n\cdot \left(1+\frac{n_{p}}{1-n_{p}}\right)$$

The addition of a second structural parameter made it possible to create a cubic exponential function based on the two mentioned structural parameters. The basic form of the function, Equation (12), is the same for all studied properties (generally denoted by *z*). The subscripts *c* and *m* are valid for composite and non-porous matrices. The values of the exponent *b* and *c* were chosen according to the best fitting. The possibility of simplifying Equation (12) is advantageous if the material is less complex [32].
(12)$$z_{c}=z_{m}\cdot n_{p}^{b}\cdot {\left(1-v_{f}\right)}^{c}$$

The values of the exponents *b* and *c* from Equation (12) can be interpolated using logarithmic functions to obtain discrete relationships valid for various mechanical properties. This shows that reality is more complex than in Equations (1)–(9) with exponents as constants varying with the type of material. The study [32] was based on only one type of filler (waste rubber) and 10 types of polyurethane matrices. The obtained relationships derived from Equation (12) valid for *E* (13), *σ_Fmax_* (14), *ε_Fmax_* (15), and *A_Fmax_* (16) are presented here, as they form part of the relations proposed by us for reinforced filled systems.
(13)$$E_{c}=E_{m}\cdot n_{p}^{d+e\cdot \ln E_{m}}\cdot {\left(1-v_{f}\right)}^{f+g\cdot \ln\left(E_{m}\cdot \delta \right)}$$
(14)$$\sigma_{c,F\max}=\sigma_{m,F\max}\cdot {n_{p}}^{d+e\cdot \ln\left(\frac{\sigma_{m,F\max}}{S_{{}_{m,rel}}}\right)}\cdot {\left(1-v_{f}\right)}^{f+g\cdot \ln\left(\frac{\sigma_{m,F\max}\cdot \delta }{S_{m,rel}}\right)}$$
(15)$$\varepsilon_{c,F\max}=\varepsilon_{m,F\max}\cdot n_{p}^{d+e\cdot \ln\varepsilon_{m,F\max}}\cdot {\left(1-v_{f}\right)}^{f+g\cdot \ln\varepsilon_{m,F\max}}$$
(16)$$A_{c,F\max}=A_{m,F\max}\cdot n_{p}^{d+e\cdot \ln\sigma_{m,F\max}}\cdot {\left(1-v_{f}\right)}^{f+g\cdot \ln\varepsilon_{m,F\max}}$$

The letters *d*, *e*, *f*, and *g* are numbers typical for each exact equation, *δ* is the expected polarity (should relate to adhesion) of the polyurethane matrix based on the OH/NCO rate before curing, and *S_m,rel_* is a dimensionless parameter related to an area lying below the tensile curve of matrix between the beginning and ultimate strength achievement. *S_m,rel_* can be calculated according to Equation (17) associated with *A_Fmax_*.
(17)$$S_{m,rel}=\frac{A_{m,F\max}}{\sigma_{m,F\max}\cdot \varepsilon_{m,F\max}}$$

To calculate the tensile modulus of elasticity for non-porous reinforced composites, it is necessary to define a mixing rule. It is an important pillar of our inspiration for creating new proposed relationships valid for porous reinforced and filled composites. The mixing rule is shown by Equation (18).
(18)$$E_{c}=E_{m}\cdot v_{m}+\eta \cdot \left(E_{r}\cdot v_{r}\right)$$
where the labels *E_c_*, *E_m_*, and *v_m_* have the same meaning as in the previous equations, *E_r_* and *v_r_* are the tensile modulus and volume fraction of the reinforcement, respectively. The last designation *η* stands for reinforcement efficiency and it is related to the adhesion between the matrix and fibers.

The material used in this study contains randomly shaped particles (and consequently irregular shaped pores) and this mixture acts as a composite matrix in our reinforced material. The reinforcement is macroscopic compared to sources in the literature [23,25,27]. Our approach builds on previous works [31,32] dedicated to filled porous systems and it follows them directly through selected matrices and fillers as well as computations. In this work, a low-volume fraction of polyethylene terephthalate (PET) monofilaments was newly added in the direction of tensile loading to investigate changes in material behavior and to extend the proposed relationships from the description of filled to filled and reinforced materials.

Our global goal is to obtain the data from composites with different compositions and observe how the shapes of relationships and their members will change compared to different parameters of individual components. In this work, only one type of filler and a small content of one type of reinforcement was used. Thus, this study makes it possible to observe changes in the proposed relationship parameters with respect to matrix change, not the filler or reinforcement. Nevertheless, the goal of this work is a great step of incorporating the reinforcement into the previously filled material and its simpler relationships and observing the changes by the addition of the mixing rule and its correction and basic description of new parameters.

## 2. Materials and Methods

### 2.1. Materials

All polyurethane matrices were based on Unixin PU4223CS pre-polymer (Lear, s.r.o., Brno, Czech Republic) composed of methylene diphenyl diisocyanate (M_n_ = 690 g·mol^−1^, 6.9 wt.% of isocyanate groups). Glycerol (Penta Chemicals, Chrudim, Czech Republic) and castor oil (Fichema, Brno, Czech Republic) served as curing agents used alone or together. Linseed oil (Fichema, Brno, Czech Republic) was used as a plasticizer in one of in used matrices. Dibutyl-tin dilaurate (Lear, Brno, Czech Republic) was used as a catalyst.

Randomly shaped waste rubber from car tires was used as a filler and was supplied by RPG Recycling s.r.o. (Uherský Brod, Czech Republic). The particle size distribution of the used filler was characterized by a laser analyzer HELOS (H2568) and RODOS and is shown in Figure 1. The rubber density of 1.18 g·cm^−3^ was measured by the pycnometer method.

PET monofilaments were used as reinforcement. Their cross-section was rectangular and the average value of area including deviations was 0.45 ± 0.08 mm^2^. The length corresponding to the prepared slabs was 120 ± 3 mm.

### 2.2. Sample Preparation

First, the pre-polymer was mixed by hand with the curing agent(s), catalyst, and plasticizer. The rubber filler (20–90 vol.%, if porosity is neglected) was then added to the liquid mixture and carefully homogenized. This mixture was partially filled into molds covered with polyethylene foil for better separation. Three PET monofilaments were added giving approximately 0.4–0.5 vol. % of porous material and 0.5–0.9 vol. % if the porosity is neglected. The orientation of monofilaments was longitudinal with respect to the tensile load. The remaining amount of the filled system was added to the mold to achieve the final dimensions of 120 × 24 × 12 mm. Finally, the samples were pressed by hand. Curing was carried out under ambient conditions for four days. The prepared samples were weighted, and their dimensions were measured to calculate their porosity. In all cases, only a very small volume fraction of reinforcement was used, so a simplified calculation of sample porosity based on knowledge of phase densities and volume fractions of matrix and filler with a combination of sample dimensions and the presence of PET monofilaments was neglected. PET density was therefore not measured.

### 2.3. Used Matrices

The designation of matrices is based on their composition and is: P_95_-G_5_; P_80_-G_20_; P_85_-G_5_-CO_10_; P_65_-CO_35_ and P_49_-CO_26_-LO_25_, where P is polyurethane pre-polymer PU4223CS, G is glycerol, CO is castor oil and LO is linseed oil. The numbers in the indexes mean the vol. % of the component. Dibutyltin dilaurate was used as a catalyst and its content was 0.03 wt. % (in case of any presence of glycerol) or 0.1 wt.% (in the absence of glycerol) compared to the weight of the pre-polymer. The OH/NCO molar ratio before curing (*δ*) presented an interesting range and depended on the type(s) and content of curing agent(s) and varied from 0.87 (matrices without glycerol) through 1.22 (P_95_-G_5_) and 1.55 (both curing agents) to 5.79 (P_80_-G_20_). The mechanical properties (obtained by fitting in [32]) of the hypothetical non-porous matrices are mentioned in Table 1. Densities of non-porous matrices (in g·cm^−3^) are 1.03 (P_49_-CO_26_-LO_25_), 1.04 (P_65_-CO_35_), 1.05 (P_85_-G_5_-CO_10_), 1.12 (P_95_-G_5_) and 1.15 (P_80_-G_20_).

### 2.4. Characterization Methods

The density of the rubbery filler was measured using a pycnometer in acetone. The density of porous matrices was obtained directly from the weight (8–18 g, by analytical balance) divided by the volume of the sample corresponding to the volume difference of ethanol added to a narrow-necked container in the presence/absence of the sample (measurement repeated with three samples) [32].

The porosity of the matrix leading to the calculation of the hypothetical density of the non-porous matrix was determined using confocal laser scanning microscopy (Lext OLS 3000, Olympus). The value used was the average of 10 measurements. The measurement was performed as the rate analysis of void cross-section sum in fracture area in optical mode [32].

The tensile test was performed using a universal static material testing machine (ZWICK Z010 ROELL). The strain rate was 30 mm∙min^−1^. Tensile modulus (*E*), ultimate strength (*σ_Fmax_*), ultimate strain (*ε_Fmax_*), and specific energy needed for ultimate strength achievement (*A_Fmax_*) were determined from the measurement. The tensile modulus was determined from the linear part of the tensile curve in the strain range of 0.05–0.25 % [32]. Each sample series included five tested samples to achieve good reproducibility of results.

## 3. Results and Discussion

As was mentioned in the introduction, we use the macroscopic point of view when we connect the structure(composition) with the mechanical properties due to the random location of pores. So, we consider our materials as isotropic except for the reinforcement which makes our calculations quite easy from the mathematical point of view. However, there are in the article has a lot of symbols and parameters and many calculation steps following one another. This situation requires a good explanation for a better understanding of the calculation process presented in this work. First, there is Appendix A serves as a list of symbols. The following Appendix A describes the origin and meaning of members in the equation dedicated to elastic modulus calculation. This property serves as an example because the calculation method for all properties is the same with a very low number of little differences. The calculation process including the work with the data is shown also in Appendix A serving as a diagram for better understanding. A flow chart showing the research evolution is shown in Appendix A. The elastic modulus serves as an example property in the diagrams. Appendix A, and Appendix A are placed in the Appendix A.

Matrices are the binding base of our materials. Their properties obtained by fitting = belonging to hypothetical non-porous matrices (detail in [32]) are listed in Table 1. The data from Table 1 were used for further calculations. The mechanical properties of rubber filler were unknown (only available in particulate form), but it did not matter, because only one type of filler was used.

In addition to the matrices, PET monofilaments were also subjected to tensile testing with resulting values obtained from five measurements with regard to their accuracy and deviations *E* = 2700 ± 700 MPa, *σ_Fmax_* = 189 ± 17 MPa, *ε_Fmax_* = 0.38 ± 0.07 and *A_Fmax_* = (54 ± 6) × 10^3^ kJ·m^−3^, respectively. However, the calculation (further in Equations (21)–(23)) needs only one number, therefore exact averages were used—*E* = 2729 MPa, *σ_Fmax_* = 189.53 MPa, *ε_Fmax_* = 0.375 except *A_Fmax_*, respectively, i.e., not needed for further calculations. The numbers are quite high because the calculations required to keep the same unit in all of the calculations and PET is very different from used PUR matrices used in the case of mechanical property values.

The mixing rule in Equation (18) used for the tensile modulus is very similar to our proposed equations. The form of the proposed relationship is represented by Equation (19) found for the tensile modulus, while the other equations valid for the other properties are analogous.
(19)$$E_{cr}=\eta_{E}\cdot \left(E_{c}\cdot \left(1-v_{r}\right)+E_{r}\cdot v_{r}\right)$$

*η_E_* is the correction parameter. The subscript *_E_* shows that the correction parameter in this case belongs to the *E* calculation and can be replaced by different properties. The subscripts belonging to *E* values indicated how complex the system the values are valid for. The subscripts symbols *m*, *c*, *r*, and *cr* are valid for matrix, filled composite, reinforcement, and reinforced filled composite. The symbol *v_r_* is the volume fraction of reinforcement with neglect of porosity. The relationship cannot be successfully fitted without porosity neglecting in member *v_r_*! It is possible to express the correction parameter on the left side of the equation after the *E_c_* calculation (Equation (20)).
(20)$$\eta_{E}=\frac{E_{cr}}{E_{c}\cdot \left(1-v_{r}\right)+E_{r}\cdot v_{r}}$$
where *E_r_* is the average value from the measured values of PET monofilaments. *E_cr_* is the measured value of the tensile modulus of a selected porous-filled reinforced composite. The member *E_c_* can be calculated according to Equation (13) where *E_m_* values were derived from Table 1. The parameters *d*, *e*, *f*, and *g* were taken from [32]. The obtained *η_E_* values for different rubber rates (the number of PET monofilaments was constant but not their volume fraction) were fitted with the expression in Equation (21).
(21)$$\eta_{E}=h_{E}\cdot n_{p}^{i}\cdot {\left(1-v_{f}\right)}^{j}$$

The parameters *i* and *j* were selected to provide the best fit. The parameter *h_E_* is the slope of obtained linear dependence passing through the beginning. The subscript varies by property—here *E*! The primary fitting was repeated for composites with different base PUR matrices to obtain different *h_E_*, *i*, and *j* values (see A part of Figure 2). The given parameters were then adjusted (see Figure 3, Figure 4 and Figure 5). For the other observed mechanical properties, the same approach as for *E* (obtaining the parameters *h_z_*, *i*, and *j* parameters (Figure 2) and their further interpolation (Figure 3, Figure 4 and Figure 5) valid for *E* and *ε*). It is convenient to show the obtained equations (Equations (22)–(24)) valid for other properties (*σ_Fmax_*, *ε_Fmax_*, and *A_Fmax_*) corresponding to Equation (19) valid for *E*.
(22)$$\sigma_{cr,F\max}=\eta_{\sigma }\cdot \left(\sigma_{c,F\max}\cdot \left(1-v_{r}\right)+\sigma_{r,F\max}\cdot v_{r}\right)$$
(23)$$\varepsilon_{cr,F\max}=\eta_{\varepsilon }\cdot \left(\varepsilon_{c,F\max}\cdot \left(1-v_{r}\right)+\varepsilon_{r,F\max}\cdot v_{r}\right)$$
(24)$$A_{cr,F\max}=\eta_{A}\cdot \left(A_{c,F\max}\cdot \left(1-v_{r}\right)+A_{r,F\max}\cdot v_{r}\right)$$
where the values of *σ_c,Fmax_*, *ε_c,Fmax_*, and *A_c,F,max_* were calculated by Equations (14)–(16) shown in the introduction.

The coefficients *h_z_*, *i*, and *j* obtained by fitting the primary data were then submitted to logarithmic interpolation. In the case of *i* and *j*, it was similar to cases of *b* and *c* valid for unreinforced materials [32]. Instead of *b* and *c*, there were two possibilities for how to accomplish the interpolation. The interpolation was completed by the same values as for *b* and *c* (values of quantity valid for hypothetic non-porous polyurethane matrix) or by the same mean as for *b* and *c* carried out by using the values of slope—here *h_z_*. The interpolation leads to proposed Equations (25)–(33). Partial coefficients were labeled according to the next part of the alphabet *k*, *l*, *m*, *n*, *o*, and *p*. The values of *m* and *n*, resp. *o* and *p* were various in the case of interpolation of *i* and *j* in left and right versions of relationships (26), (27), (29), (30), (32) and (33). In each case (a combination of quantity and coefficient), only five points were obtained according to a number of polyurethane matrices. The fitting of coefficients *h_z_*, *i*, and *j* used to describe *E* and *ε_Fmax_* is shown in Figure 3, Figure 4 and Figure 5.

η_E_:(25)$$h_{E}=k+l\cdot \ln E_{m}$$
(26)$$i=m+n\cdot \ln E_{m}\;\mathrm{resp}.\;i=m+n\cdot \ln h_{E}$$
(27)$$j=o+p\cdot \ln\left(E_{m}\cdot \delta \right)\;\mathrm{resp}.\;j=o+p\cdot \ln\left(h_{E}\cdot \delta \right)$$

*η_σ_*: *E_m_* from (25)–(27) is substituted by $\frac{\sigma_{m,F\max}}{S_{m,rel}}$ and *h_E_* by *h_σ_*.

ηε:(28)$$h_{\varepsilon }=k+l\cdot \ln\varepsilon_{m,F\max}$$
(29)$$i=m+n\cdot \ln\varepsilon_{m,F\max}\;\mathrm{resp}.\;i=m+n\cdot \ln h_{\varepsilon }$$
(30)$$j=o+p\cdot \ln\varepsilon_{m,F\max}\;\mathrm{resp}.\;j=o+p\cdot \ln h_{\varepsilon }$$

η_A_:(31)$$h_{A}=k+l\cdot \ln A_{m,F\max}$$
(32)$$i=m+n\cdot \ln\sigma_{m,F\max}\;\mathrm{resp}.\;i=m+n\cdot \ln h_{\sigma }$$
(33)$$j=o+p\cdot \ln\varepsilon_{m,F\max}\;\mathrm{resp}.\;j=o+p\cdot \ln h_{\varepsilon }$$

Of note, the parameters *m* and *n* in Equation (32) are different from the parameters in the equation valid for *η_σ_* and the parameters *o* and *p* in Equation (33) are different from parameters *o* and *p* in Equation (30), although the equations appear to be the same.

It should be mentioned that adhesion plays a very important role in the mechanical behavior of each any reinforced (even filled) composite [33]. In this study, only a small volume fraction of reinforcement was added, and the rest of the material remained the same as in [32]. It is assumed, that the adhesion between the reinforcement and the composite matrix depended primarily only on the structure of the composite matrix. The calculations dedicated the correction parameter of the mixing rule on the structure of the composite matrix. This can be seen, e.g., for example in the shape of the cubic exponential equations shape (the same for the properties of filled materials in [32]) and the use of the same structural parameters—interspace filling and interspace volume. This is the first published result of this approach, and this result is rather crude and is probably applicable only for materials with a low volume fraction of reinforcement. It is anticipated that research into more types of porous-filled and reinforced materials will be required to refine the relationships. It will be necessary to include also materials containing a higher volume fraction of reinforcement.

Finally, the chosen approach to the structure-property relationship appears to be mathematically simple and uses a practical macroscopic viewpoint for systems that seem macroscopically homogeneous. However, the characterization of simple components of such a complicated system as the porous-filled and reinforced composite is very advantageous and could be emphasized in further research. The relationship can be moreover conveniently simplified when the described system is simpler due to the form of an equation consisting of several parts corresponding to the material composition. It must be added that a macroscopically homogeneous composition and thus the mechanical behavior (except for reinforcement) of the material is necessary and expected. Now, only the reinforcement orientation in the direction of loading is considered.

## 4. Conclusions

The proposed study can be an important way to describe the mechanical behavior of porous-filled and reinforced composites. The presented Equations depend on a special form of mixing rule combined with a relationship describing the porous-filled composite serving as a matrix. The mixing rule cannot be fully applicable without the correction parameter *η*. This parameter can be subjected to fitting and further interpolation of some terms in the obtained functions. The process is very similar to the case of properties (e.g., tensile modulus) of a filled porous composite without reinforcement. The first step is fitting by a cubic exponential function containing suitable structural parameters, and the second step is trying to find the meaning of the exponents (interpolation by matrix properties).

The offered relationships were tested on a relatively limited number of sample compositions including only one type of filler and one type of reinforcement. However, the proposed relationships promise the possibility of connecting the structure (composition) with the mechanical behavior of porous material with varying complexity.

This claim requires further research to link the exact relationships. It should also include more different types of materials, especially higher volume fractions and orientation of reinforcement. Further research should be more concerned with the adhesion of filler and reinforcement to the matrix as opposed to an assumption based on chemical composition as in our article.

## Figures and Tables

**Figure 1 polymers-14-04390-f001:**
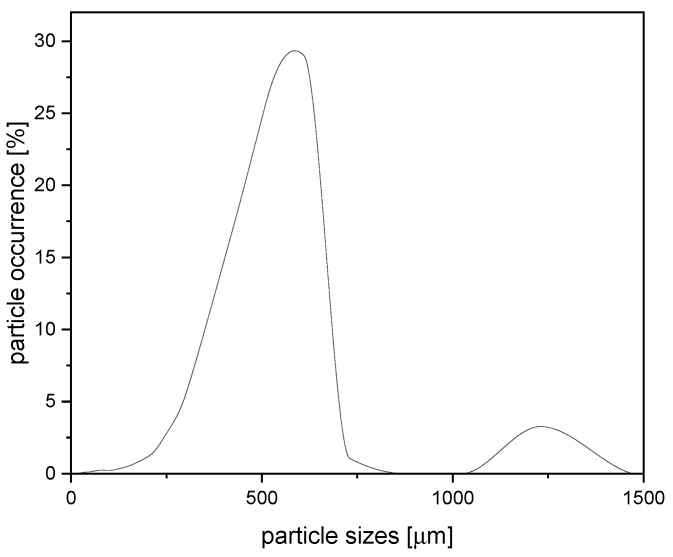
Particles size distribution of rubber filler. Reprinted by permission from Springer Nature Customer Service Centre GmbH: Springer Nature, SN Applied Sciences, A new approach to the structure-properties relationship evaluation for porous polymer composites, M. Cerny et al., 2020, doi:10.1007/s42452-020-2479-8 [SN Appl. Sci.], https://www.springer.com/journal/42452, accessed on 23 August 2022.

**Figure 2 polymers-14-04390-f002:**
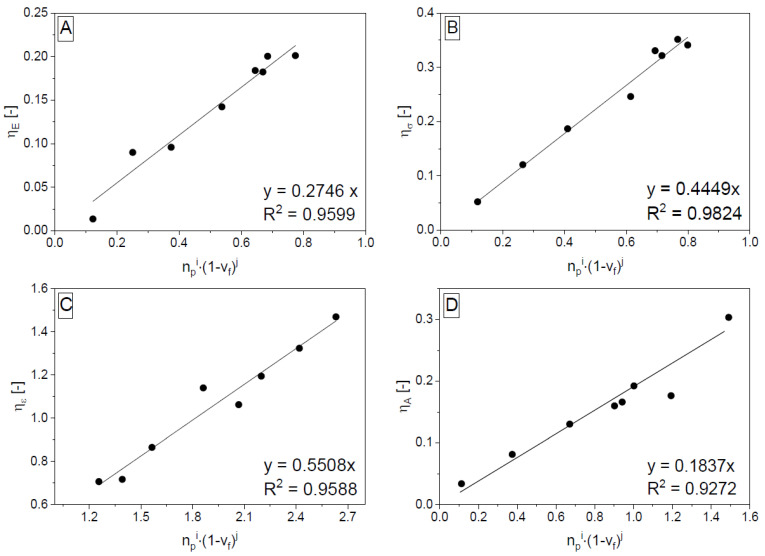
Primary data fitting valid for porous composites: P_65_-CO_35_ matrix filled with ground rubber in different ratios reinforced by PET monofilaments (3 in each tested specimen). Graphs from (**A**) to (**D**): fitting of mixing rule correction parameters (*η_E_*, *η_σ_*, *η_ε_*, and *η_A_*) according to powered structural parameters *n_p_* and 1 − *v_f_* leading to Equation (21) valid for *η_E_* calculation, and analogous equations valid for *η_σ_*, *η_ε_*, and *η_A_* calculations. The mixing rule can be seen in Equations (19) and (20)—versions for *E*, (22)—for *σ_Fmax_*, (23)—for *ε_Fmax_*, and (24)—for *A_Fmax_*.

**Figure 3 polymers-14-04390-f003:**
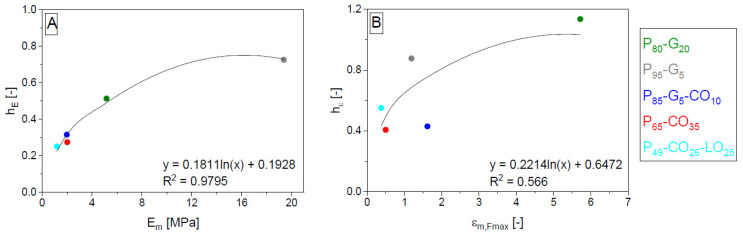
Dependences corresponding to relations (25) (**A**) and (28) (**B**). The symbol *h* with subscript labeling the property is the slope of dependence describing the correction parameter *η* in the mixing rule. The symbol *h* is here described by polyurethane matrix properties. Matrices designation is composed of components labeled P (polyurethane pre-polymer), G (glycerol), CO (castor oil), and LO (linseed oil). Numbers (subscripts) mean vol. % of matrix components.

**Figure 4 polymers-14-04390-f004:**
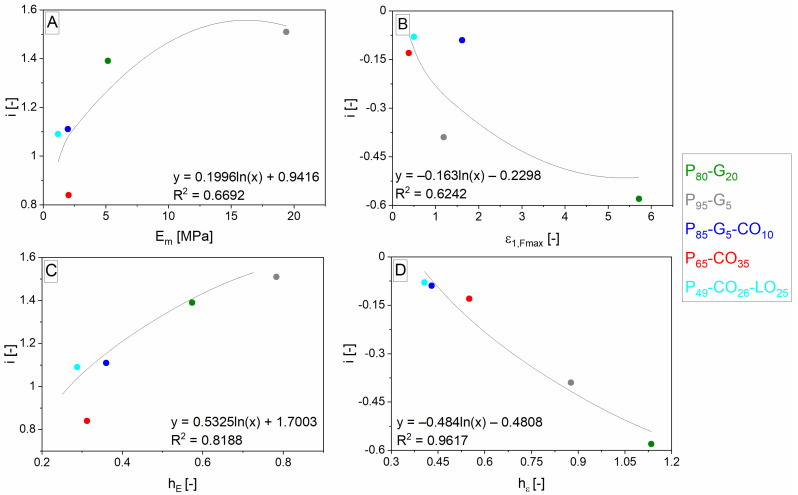
Dependences corresponding to the left (**A**) and right (**C**) side of relationship (26) resp. the left (**B**) and right (**D**) sides of the relationship (29). The symbol *i* is one from two exponents in different dependencies describing the mixing rule correction parameters *η* and depends on the polyurethane matrix properties or slope *h* of the same dependencies where they are used. The values of *η*, *h* and *i* vary according to described properties. Matrices designation is composed of components labeled P (polyurethane prepolymer), G (glycerol), CO (castor oil), and LO (linseed oil). Numbers (subscripts) mean vol. % of matrix components.

**Figure 5 polymers-14-04390-f005:**
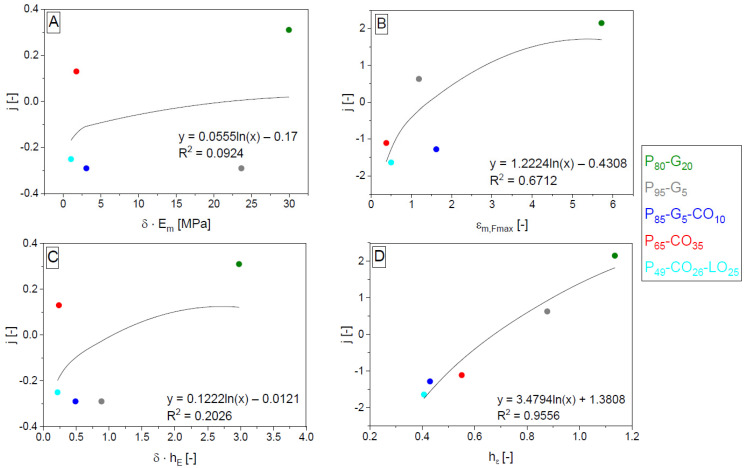
Dependences corresponding to the left (**A**) and right (**C**) side of relationship (27) resp. the left (**B**) and right (**D**) sides of the relationship (30). The symbol j is one of two exponents in different dependences describing the mixing rule correction parameters *η* and depends on the polyurethane matrix properties or slope *h* of the same dependencies where they are used. The values of *η*, *h*, and *j* differ according to described properties. Matrices designation is composed of components labeled P (polyurethane prepolymer), G (glycerol), CO (castor oil), and LO (linseed oil). Numbers (subscripts) mean vol. % of matrix components.

**Table 1 polymers-14-04390-t001:** Mechanical properties of non-porous matrices obtained by fitting of filled composites mechanical properties according to Equation (12) used for calculations as *E_m_*, *σ_m,Fmax_*, *ε_m,Fmax_*, and *A_m,Fmax_* (according to property) in Equations (13)–(17) and (25)–(33). Data come from the article [32] *. Matrices designation is composed of components labeled P (polyurethane pre-polymer), G (glycerol), CO (castor oil), and LO (linseed oil). Numbers (subscripts) mean vol. % of matrix components.

Designation	E (MPa)	σ_Fmax_ (MPa)	ε_Fmax_ (-)	A_Fmax_ (kJ·m^−3^)
P_95_-G_5_	19.38	5.17	1.19	3377
P_80_-G_20_	5.17	1.49	5.72	6920
P_85_-G_5_-CO_10_	1.98	1.02	1.62	1004
P_65_-CO_35_	2.02	0.42	0.38	111
P_49_-CO_26_-LO_25_	1.19	0.29	0.50	83

* Reprinted by permission from Springer Nature Customer Service Centre GmbH: Springer Nature, SN Applied Sciences, A new approach to the structure-properties relationship evaluation for porous polymer composites, M. Cerny et al., 2020, doi:10.1007/s42452-020-2479-8 [SN Appl. Sci.], https://www.springer.com/journal/42452, accessed on 23 August 2022.

## Data Availability

The data presented in this study are available on request from the corresponding author.

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
