# Peer review of "A New Approach to the Structure-Properties Relationships Determination for Porous Filled Reinforced Materials"

_polymers, 2022, doi:10.3390/polym14204390_

Round 1

Reviewer 1 Report

The manuscript entitled "A new approach to the structure-properties relationships determination for porous filled reinforced materials" provided a new mathematical approach to the relationship between mechanical properties, composition, and structure for porous filled reinforced composites based on polyurethane matrices. The manuscript presents interesting results. Moreover, the explanation of the phenomena observed is quite satisfactory whereas the text is fairly well written and exhibits an element of originality. Thus, I recommend it for publication after small changes. The suggestions are as follows:

1. The abstract should show a brief introduction to the importance of structure-properties relationships determination before its objective to keep the abstract comprehensible to scientists working outside the topic of the paper. So, the abstract must be able to stand alone. Moreover, the authors should add more important results in this section as well. The abbreviations must be defined at their first mention in the abstract itself (PUR).

2. Introduction needs to be improved to show better the work's originality and importance. Moreover, it seems that the novelty of the work with respect to the literature, and the state of the art are not well highlighted in the paper. Also, a clear statement on what was intended to be investigated (expectations, methods of characterization, aims, hypotheses) would be helpful (line 108-113 is quite vague).

3. Let the table and figure legends be more informative (especially abbreviations and sample codes) to avoid the readers getting back to the methodology section to understand what we can see in the results.

4. There are some errors regarding the references in the text e.g. lines 186, 256, and 262. Please, correct them.

Reviewer 2 Report

This paper developed a new model to find out the structure-properties relationships for porous-filled reinforced materials. This paper can be accepted after minor revision.

1. Some editing errors, like lines 164-167, line 186, Line 256, Line 262, subchapter number in line 135, Line 150, and line 168.

2. Line 37, the authors focus on the exponential shapes. The authors can give a brief introduction about different mathematical models and highlight the advantage of exponential one.

3. A nomenclature including all parameters is necessary.

4. The authors mentioned data in Table 1 and in Line 192. In which equations are these data used?

5. The calculation procedure is not very clear. A flow chart may be helpful for understanding. 

Reviewer 3 Report

1.      Its current title's uppercase and lowercase should be updated according to MDPI format.

2.      Please include all of the author’s emails after affiliation with name initials, except for the corresponding author based on MDPI format.

3.      The abstract section should be enhanced to include quantitative data.

4.      Please add the abstract's "take-home" message, the current form was insufficient.

5.      Make the each of keywords with lowercase font following MDPI format, revise it.

6.      Describe the novelty of the article made by the author? From the results of my evaluation, it seems that many similar published works adequately explain what you have raised in the current manuscript, even the author mention it as “new approach”, nothing cutting insight in the porous materials research area. If there is something others really new in this manuscript, please highlight it more clearly in the introduction section.

7.      Previous studies must be explained in the introductory part, including their work, innovation, and limits, to demonstrate the research gaps that will be filled in the current study.

8.      Revise “Error! Reference source not found”.

9.      Why the present study studied with mathematical equation? It would be interesting and improve the article quality by additional research in from of experimental testing and computational simulation. The author should explain the explanation structurally. The potential adopting computational simulation for studied materials related also should be addressed by authors since it several advantages such as faster results and approximate in the real condition compared to analytical solution. It is a vital topic that authors must provide in the introduction and/or discussion section. Additionally, the suggested reverence should be taken to substantiate this explanation as follows: Ammarullah, M. I.; Santoso, G.; Sugiharto, S.; Supriyono, T.; Kurdi, O.; Tauviqirrahman, M.; Winarni, T. I.; Jamari, J. Tresca Stress Study of CoCrMo-on-CoCrMo Bearings Based on Body Mass Index Using 2D Computational Model. Jurnal Tribologi 2022, 33, 31–8. https://jurnaltribologi.mytribos.org/v33/JT-33-31-38.pdf

10.   To help the reader grasp the study's workflow more easily, the authors could include more visuals to the materials and methods section in the form of figures rather than sticking with the text that now predominates.

11.   A comparative assessment with similar previous research is required.

12.   Before moving on to the conclusion section, the present study's limitation must be added at end of the discussion section.

13.   Mention further research in the conclusion section.

14.   The reference should be enriched with literature from the last five years. Literature published by MDPI is strongly recommended.

15.   The authors occasionally created paragraphs in the entire document that were just one or two phrases long, which made the explanation difficult to understand. To make their explanation into a longer, more thorough paragraph, the authors should expand it. It is advised to use at least three sentences in a paragraph, with one serving as the primary sentence and the others as supporting phrases.

16.   Due to grammatical and language issues, the authors need to proofread the present work. This problem would use MDPI English editing service.

17.   Check to see if the authors followed the MDPI format exactly, then modify the current form and double-check it together with the other issues that have been noted.

Round 2

Reviewer 3 Report

Reviewers greatly appreciate the efforts that have been made by the author to improve the quality of their articles after peer review. I reread the author's manuscript and further reviewed the changes made along with the responses from previous reviewers' comments. Unfortunately, the authors failed to make some of the substantial improvements they should have made making this article not of decent quality with biased, not cutting-edge updates on the research topic outlined. In addition, the author also failed to address the previous reviewer's comments, especially on comments number 6, 7, and 9.With all due respect, the reviewer opposed this article to be published and must be rejected. Thank you very much for the opportunity to read the author's current work.

Round 3

Reviewer 3 Report

What the author did was not sufficient to respond to the previous reviewer's doubts because the novelty was not strong and the explanation was not satisfactory. Reviewers judged this work to be of low quality, making it no significant contribution. The reviewer's recommendation is to reject this manuscript for publication. Reviewers thank you for the opportunity and appreciate the author's efforts.
